# Optimum Angle of Force Production Temporarily Changes Due to Growth in Male Adolescence

**DOI:** 10.3390/children8010020

**Published:** 2021-01-03

**Authors:** Junya Saeki, Satoshi Iizuka, Hiroaki Sekino, Ayahiro Suzuki, Toshihiro Maemichi, Suguru Torii

**Affiliations:** 1Faculty of Sport Sciences, Waseda University, 2-579-15 Mikajima, Tokorozawa, Saitama 359-1192, Japan; anporian@gmail.com (S.I.); shunto@waseda.jp (S.T.); 2Japan Society for the Promotion of Science, 5-3-1 Kojimachi, Chiyoda-ku, Tokyo 102-0083, Japan; 3Graduate School of Sport Sciences, Waseda University, 2-579-15 Mikajima, Tokorozawa, Saitama 359-1192, Japan; ry52112049@gmail.com (H.S.); sumanana-107@akane.waseda.jp (A.S.); t.m.waseda@ruri.waseda.jp (T.M.)

**Keywords:** force–angle relationship, isokinetic muscle strength, muscle–tendon unit, maximal voluntary contraction, growth spurt, children

## Abstract

The peak increase in lean mass in adolescents is delayed from peak height velocity (PHV), and muscle flexibility temporarily decreases as bones grow. If the decrease in muscle flexibility is caused by muscle elongation, the relationship between the exerted torque and the joint angle could change in adolescents. The purpose of this study was to investigate the change in the optimum angle of force production due to growth. Eighty-eight healthy boys were recruited for this study. Isokinetic knee extension muscle strength of the dominant leg was recorded. The outcome variable was the knee flexion angle when maximal knee extension torque was produced (optimum angle). The age at which PHV occurred was estimated from subjects’ height history. We calculated the difference between the age at measurement and the expected age of PHV (growth age). A regression analysis was performed with the optimal angle of force exertion as the dependent variable and the growth age as the independent variable. Then, a polynomial formula with the lowest *p*-value was obtained. A significant cubic regression was obtained between optimum angle and growth age. The results suggest that the optimum angle of force production temporarily changes in male adolescence.

## 1. Introduction

The incidence of sports injuries in adolescents increases until 15 to 16 years of age and decreases thereafter [1,2]. The cause has been considered to be influenced by bone mineral density (BMD) because the whole body BMD of the adolescent decreases when peak bone length increases, and many distal radius fractures occur at this time [3]. However, while whole body BMD recovers after its lowest point at around 13 years, also the time of peak height velocity (PHV) [3], the incidence of sports injury increases until 15 to 16 years of age [1]. Thus, it is not possible to explain the cause of the high incidence of sports injuries in adolescents from BMD only.

The lean mass peak increase in adolescents is delayed from PHV [4], and muscle flexibility temporarily decreases with increasing bone length [5,6]. A previous study on the tibialis anterior of rabbits showed that the exerted force changed the muscle elongation when the tibia was cut and the bone was torn in the longitudinal direction [7]. Because muscle length is changed by the joint angle [8], the exerted muscle force is changed by a joint angle (force–angle relationship) [9]. A previous study in vivo reported that the peak torque of the isokinetic contraction did not change, but the peak angle was changed by a chronic static stretching program [10]. As such, it was suggested that the force–angle relationship may be related to the change of muscle flexibility. Considering the above background, the relationships between exerted torque and joint angle may change during the muscle elongation period when bone growth precedes muscle growth. If the force–angle relationship temporarily changes in an adolescent, it could affect body control and the occurrence of sports injuries. However, change in the force–angle relationship has not been investigated.

To prevent sports injuries in adolescents, further understanding of the force production characteristics is necessary. Therefore, the purpose of this study was to investigate the relationship between the optimum angle of force production and growth age. The muscle flexibility temporarily decreases during adolescence [5,6], which could lead to muscle elongation and a temporary change in the force–angle relationship. We hypothesized that the optimum angle of force production changes curvilinearly during adolescence.

## 2. Materials and Methods

### 2.1. Subjects

Eighty-eight healthy junior high school boys (age: 13.6 ± 1.0 years, height: 157.9 ± 9.2 cm, body mass: 47.1 ± 8.8 kg; means ± SDs) were recruited in this study. The inclusion criteria were male junior high school soccer players who participated in a medical examination in their team. Exclusion criteria included history of lower extremity orthopedic surgeries or orthopedic disorders in the lower extremities on measurement day. Written informed consent was obtained from parents of the subjects prior to participation. This study was approved by an institutional human research ethics committee (2013-167) and was carried out in accordance with the declaration of Helsinki.

### 2.2. Measurement of the Optimum Angle

Isokinetic knee extension muscle strength of the dominant leg was recorded using an electric dynamometer (BIODEX System 3, BIODEX, Shirley, NY, USA). The subjects were seated on the dynamometer with their trunk, pelvic, and dominant thigh secured to a dynamometer by using non-elastic straps. The subjects’ dominant leg was secured to an attachment and rotation axis of the knee joint was matched with the rotation axis of the dynamometer in a drooped position of the leg. The angle at which the thigh and leg of the subject were perpendicular was set as 90° of the dynamometer. At the measurement, the maximal isokinetic contraction of the knee extension was conducted at 60°/s in the range of maximal knee flex position to maximal knee extension position. The measurement was taken 3 times. We calculated the average value of the maximal knee extension torque (peak torque). The optimal angle of the force production was calculated by imposing the obtained scatter diagram of the joint angle and the exerted torque with a cubic curve and obtaining the coordinates of the local maximum point.

### 2.3. Estimation of the Growth Age

To calculate the subjects PHV age, height history in primary school was determined by listing annual physical measurement data on a questionnaire. In addition, the subjects’ height in junior high school was recorded using a height meter. PHV age was estimated from the acquired height history using analysis software (AUXAL 3.1, Scientific Software International, Skokie, IL, USA). Finally, we calculated the difference between age at measurement and PHV, as a growth age.

### 2.4. Statistics

Descriptive data are presented as means ± SDs. Statistical analyses were performed using statistical software (SPSS Statistics 26; IBM, Chicago, IL, USA). To investigate the development of the peak torque, a linear regression analysis was performed with the peak torque as the dependent variable and the growth age as the independent variable. Subsequently, to investigate the development of the optimum angle, a regression analysis was performed with the optimal angle of force exertion as the dependent variable and the growth age as the independent variable. In these analyses, a polynomial formula with the lowest *p*-value was obtained. For all tests, the statistical significance was set at *p* < 0.05.

## 3. Results

The PHV age and the growth age were 13.3 ± 0.9 and 0.3 ± 1.5 years, respectively. A significant linear regression was obtained between the peak torque and growth age (*p* < 0.001, *R*^2^ = 0.40) (Figure 1). A significant cubic regression was obtained between optimum angle and growth age (*p* = 0.01, *R*^2^ = 0.14) (Figure 2). Coordinates of the minimal local value and maximal local value were (−1.0, 67.6) and (2.3, 74.6), respectively.

## 4. Discussion

The present study investigated the relationship between the optimum angle of force production and growth age. The main finding of the present study was that the relationship between the optimum angle and growth age was regressed by a cubic curve. To the best of our knowledge, this is the first study investigating the change in the optimum angle of force production during growth.

The PHV age was 13.3 ± 0.9 in this study and this result was the same as that reported by a previous study for Japanese subjects [11]. However, the PHV age was 0.2 to 0.8 years earlier than previous studies for Europeans and North Americans [4,12,13]. It is considered that race might influence the inter-study differences in PHV age.

A significant linear regression relationship was obtained between the peak torque and growth age. The relationship between optimum angle and growth age was significant when analyzed using the cubic regression model. These results support our hypothesis, and suggest that the optimum angle temporarily changes, despite the peak torque of force production developed with growth. The optimum angle of knee extension force production was temporarily smaller at approximately 1 year before PHV because the growth age on the minimal local value was −1.0 years. According to previous studies, it is reported that the increase in femur length of male adolescent peaks at 1.1 years before PHV [14]. The growth age when the optimum angle reaches its minimal value in this study nearly coincided with the increase in femur length in the previous study. In addition, it has been reported in other studies of the Japanese population that the flexibility of the quadriceps femoris muscle decreases in around 12-year-old individuals [5]. The optimum angle of force production in the present study was shown to be the minimum local around 12 years old when converted to age. Considering this, the increase of bone length and decrease of muscle flexibility could be related to temporary changes of the optimum angle of the knee extension muscle force production.

The results of the present study showed that the optimum angle of knee extension muscle force production temporarily decreases with growth. The muscle lengthens by an increase in bone length [7], and muscle length (joint angle) affects the force production [9,15]. Therefore, muscle elongation due to an increase of bone length could affect the force–angle relationship. In particular, because the quadriceps femoris muscle is elongated when the knee is flexed, muscle elongation due to growth is considered to cause the same change as knee flexion. Based on the above, the optimum angle of knee extension force production was considered to temporarily decrease due to growth.

The growth age of the maximum local value on the regression curve was 2.3 years. The peak of the temporary decrease in BMD was almost the same as that of PHV [3], which was earlier than the timing at which the optimum angle became the original value after the temporary change. In addition, because the average PHV age was 13.3 years, the optimum angle of force production was shown to be a maximum local at 15 to 16 years old when converted to age. In a previous study in Europeans and North Americans, the incidence of sports injuries in adolescents increased until 15 to 16 years of age and decreased after this period [1,2]. Although the PHV age in the present study was 0.2 to 0.8 years earlier than previous studies for Europeans and North Americans [4,12,13], the maximum local of the optimum was just before or almost the same as the timing when the incidence of sports injury begins to decline. Therefore, adolescent sports injuries could be affected by changes in force production characteristics due to growth.

There are a few limitations in this study. First, because the knee joint angle during force production was not measured, the joint angle in this study is different from the actual joint angle. Therefore, we do not mention the specific joint angle. However, the knee joint angle was defined using the same method in all subjects. Thus, this limitation cannot affect the result that the optimum angle of force production changes due to growth in adolescents. Second, because the range of the subjects’ age was limited, it is not clear when the change of the optimal angle begins. A previous study showed that the muscle flexibility of the quadriceps muscle decreased in boys aged 11 years [5]. In addition, there is a sex difference in the incidence of Osgood–Schlatter disease, which is related to the quadricep muscle flexibility [16,17]. Therefore, further research needs to include at least 11-year-old boys and investigate the sex difference of the optimum angle change.

## 5. Conclusions

This study investigated the change in the optimum angle of knee extension force production due to growth. The results showed that the optimal angle is temporarily changed to the loose position of the quadriceps femoris muscle at approximately the same time as the increase in femur length in a previous study, and returned to the original within a few years. This suggests that the force–angle relationship temporarily changes with muscle elongation due to skeletal growth.

## Figures and Tables

**Figure 1 children-08-00020-f001:**
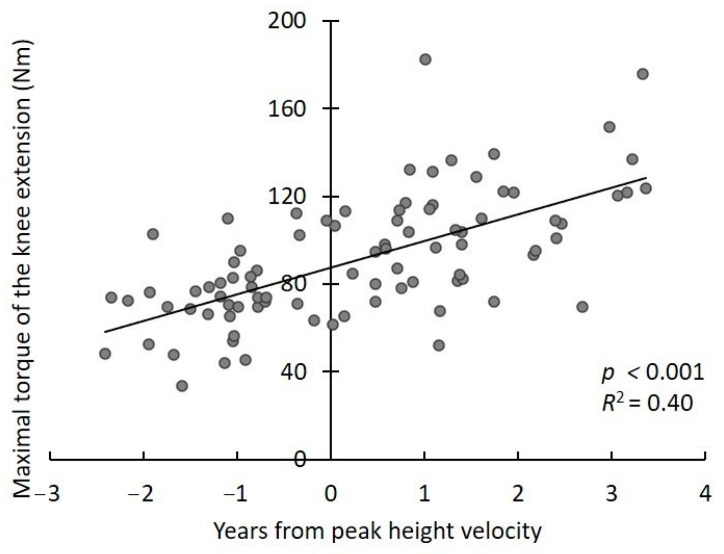
Relationships between maximal torque of knee extension and growth age.

**Figure 2 children-08-00020-f002:**
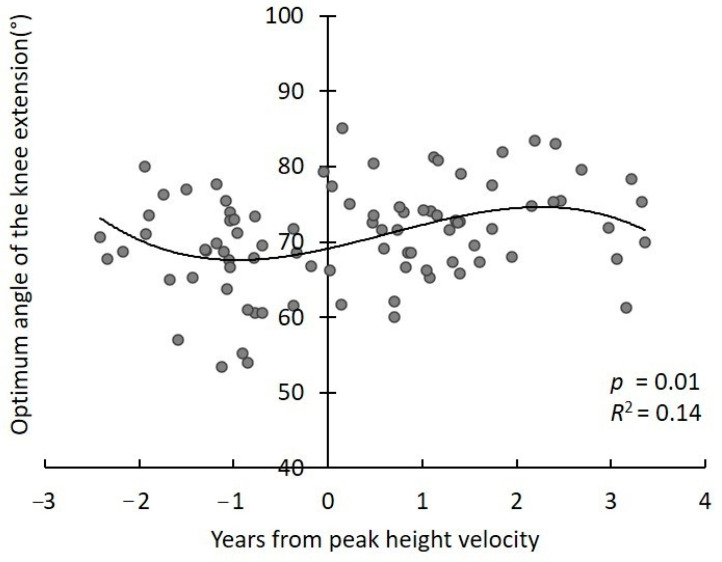
Relationships between optimum angle of the knee extension and growth age.

## Data Availability

The data presented in this study are available on request from the corresponding author.

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
