# Peer review of "Optimum Angle of Force Production Temporarily Changes Due to Growth in Male Adolescence"

_children, 2021, doi:10.3390/children8010020_

Round 1
Reviewer 1 Report
This manuscript presents a relevant topic to publish in Children, which needs some minor revisions.
In my opinion, the introduction provides adequate information and structure to set up the research questions raised in manuscript; the methodology provides sufficient detail, but that can still be an improvement; results section is sufficiently clear and precise; the discussion of results based on previous literature.
After carefully reading your manuscript, I point out some aspects that must be improved and corrected:
- I have some reservations about the title of the study because I think it should be more objective considering that it only focuses on male adolescents
- In the methodology (subjects), it should include the exclusion or inclusion criteria for sample;
- in the following sentence, the authors write"… further research needs to investigate the change of the optimum angle in younger population" (line 138-139): What ages? what age groups? both sexes? Do girls have similar results? I think the conclusion, and particularly when pointing to future studies, these questions may be better developed/substantiated.
- Some aspects of formatting should be corrected (spelling). Please, correct what it is pointed out in the body of the manuscript;
- All statistical symbols must be in italics (N, n, p, r, F ....).

Author Response
Response to Reviewer #1
Point 1:
This manuscript presents a relevant topic to publish in Children, which needs some minor revisions.
In my opinion, the introduction provides adequate information and structure to set up the research questions raised in manuscript; the methodology provides sufficient detail, but that can still be an improvement; results section is sufficiently clear and precise; the discussion of results based on previous literature.
After carefully reading your manuscript, I point out some aspects that must be improved and corrected:
Response 1:
We thank the reviewer for the constructive comments and suggestions on our manuscript. Below is a point-by-point reply to the reviewer’s comments.
Point 2:
I have some reservations about the title of the study because I think it should be more objective, considering that it only focuses on male adolescents.
Response 2:
We have added the subject information (in male adolescence) in the title.
In addition, “optimum angle changes curvilinearly” was revised to “optimum angle temporarily changes.”
Revised Title
Optimum angle of force production temporarily changes due to growth in male adolescence
Point 3:
In the methodology (subjects), it should include the exclusion or inclusion criteria for sample;
Response 3:
We have added the exclusion or inclusion criteria in the Methods section.
Revised Methods (Lines 56-59):
The inclusion criteria were male junior high school soccer players who participated in a medical examination in their team. Exclusion criteria included history of lower extremity orthopedic surgeries or orthopedic disorders in the lower extremities on measurement day.
Point 4:
In the following sentence, the authors write"… further research needs to investigate the change of the optimum angle in younger population" (line 138-139): What ages? what age groups? both sexes? Do girls have similar results? I think the conclusion, and particularly when pointing to future studies, these questions may be better developed/substantiated.
Response 4:
Since a previous study showed that the muscle flexibility of the quadriceps muscle was decreased at least in boys aged 11 years, further research needs to include at least 11 years of age for boys. In addition, since there is a sex difference in the incidence of the Osgood–Schlatter disease, which is related to quadriceps muscle flexibility, further research is needed to investigate the sex difference in the optimum angle change. We have added these descriptions in the Discussion section.
Revised Discussion (Lines 146-150):
A previous study showed that the muscle flexibility of the quadriceps muscle decreased in boys aged 11 years [5]. In addition, there is a sex difference in the incidence of Osgood–Schlatter disease, which is related to the quadriceps muscle flexibility [16, 17]. Therefore, further research needs to include at least 11 years of age for boys and investigate the sex difference of the optimum angle change.
Point 5:
Some aspects of formatting should be corrected (spelling). Please, correct what it is pointed out in the body of the manuscript.
Response 5:
As you pointed out, we have corrected the spelling in the manuscript.
Point 6:
All statistical symbols must be in italics (N, n, p, r, F ....).
Response 6:
We have changed the statistical symbols to italics.
Reviewer 2 Report
Overall comments: This study reports a valuable finding demonstrating the relationship between growth during adolescence and joint angle and knee extension torque production.
Introduction/Background: is thorough and concisely provides rationale for the study and hypothesis.
Method/Analysis: appropriate for the hypothesis.
Results/Discussion:
Line 100-101. Please clarify the point, 'because the subjects were Japanese'.
Limitation in absence of reporting specific joint angle is relevant.
Line 127-129 requires a change in wording to provide clarity and meaning to the sentence.
Conclusion: Recommend focusing the conclusion on the relationship between skeletal growth, potential change in muscle length/flexibility during the adolescent period of skeletal growth. The last sentence solely mentions adolescence and loses the specific, speculative contribution of growth and muscle length to the study and joint angle/torque. Adolescence is more vague and can encompass other contributing factors, i.e. hormones.
While not the purpose of this paper, the shortening of muscle in children with neurologic injuries during adolescence and risk of decreased range of motion may be a result of the second highest period of growth, i.e. adolescence (compared to 0-1 years of age) and lack of synergy in muscle growth. A possible extension of the meaningful application of this methodology to other populations. Not necessary to include here, simply an observation.
Author Response
Response to Reviewer #2
We thank the reviewer for the constructive comments and suggestions on our manuscript. Below is a point-by-point reply to the reviewer’s comments.
Point 1:
Overall comments: This study reports a valuable finding demonstrating the relationship between growth during adolescence and joint angle and knee extension torque production.
Introduction/Background: is thorough and concisely provides rationale for the study and hypothesis.
Response 1:
We have added the rationale for the study and hypothesis.
Revised Introduction (Lines 49-51):
The muscle flexibility temporarily decreases during adolescence [5, 6], which could lead to muscle elongation and a temporary change in the force-angle relationship.
Point 2:
Method/Analysis: appropriate for the hypothesis.
Response 2:
We are pleased to know that you find the method and analysis appropriate for the hypothesis.
Point 3:
Results/Discussion:
Line 100-101. Please clarify the point, 'because the subjects were Japanese'.
Response 3:
As you have emphasized, the statement “because the subjects were Japanese” is not clear. We have revised this paragraph accordingly.
Revised Discussion (Lines 105-108):
The PHV age was 13.3 ± 0.9 in this study, and this result was the same as that reported by a previous study for Japanese subjects [11]. However, the PHV age was 0.2 to 0.8 years earlier than previous studies for Europeans and North Americans [4, 12, 13]. It is considered that race might influence the inter-study differences in PHV age.
Point 4:
Limitation in absence of reporting specific joint angle is relevant.
Response 4:
We have added the statement of the specific joint angle in the Limitations section.
Revised Discussion
Therefore, we did not mention the specific joint angle.
Point 5:
Line 127-129 requires a change in wording to provide clarity and meaning to the sentence.
Response 5:
As you pointed out, this sentence is not clear. We have revised the entire paragraph.
Revised Discussion (Lines 130-140)
The growth age of the maximum local value on the regression curve was 2.3 years. The peak of the temporary decrease in BMD was almost the same as that of PHV [3], which was earlier than the timing at which the optimum angle became the original value after the temporary change. In addition, because the average PHV age was 13.3 years, the optimum angle of force production was shown to be a maximum local at 15 to 16 years old when converted to age. In a previous study in Europeans and North Americans, the incidence of sports injuries in adolescents increased until 15 to 16 years of age and decreased after this period [1, 2]. Although the PHV age in the present study was 0.2 to 0.8 years earlier than previous studies for Europeans and North Americans [4, 12, 13], the maximum local of the optimum was just before or almost the same as the timing when the incidence of sports injury begins to decline. Therefore, adolescent sports injuries could be affected by changes in force production characteristics due to growth.
Point 6:
Conclusion: Recommend focusing the conclusion on the relationship between skeletal growth, potential change in muscle length/flexibility during the adolescent period of skeletal growth. The last sentence solely mentions adolescence and loses the specific, speculative contribution of growth and muscle length to the study and joint angle/torque. Adolescence is more vague and can encompass other contributing factors, i.e. hormones.
Response 6:
In conclusion, we focused on skeletal growth and muscle elongation at an optimum angle.
Revised Discussion (Lines 152-156)
This study investigated the change in the optimum angle of knee extension force production due to growth. The results showed that the optimal angle is temporarily changed to the loose position of the quadriceps femoris muscle at approximately the same time as the increase in femur length in the previous study and returned to the original within a few years. This suggests that the force-angle relationship temporarily changes with muscle elongation due to skeletal growth.
Point 7:
While not the purpose of this paper, the shortening of muscle in children with neurologic injuries during adolescence and risk of decreased range of motion may be a result of the second highest period of growth, i.e. adolescence (compared to 0-1 years of age) and lack of synergy in muscle growth. A possible extension of the meaningful application of this methodology to other populations. Not necessary to include here, simply an observation.
Response 7:
Thank you for your comment. As you stated, neurologic injuries could affect the synergy in muscle growth and biodynamics. I will consider future research design and discussion bearing this perspective in mind.